# Medication Adherence Measurement in Chronic Diseases: A State-of-the-Art Review of the Literature

**DOI:** 10.3390/nursrep15100370

**Published:** 2025-10-16

**Authors:** Jacqueline Dunbar-Jacob, Jian Zhao

**Affiliations:** 1School of Nursing, University of Pittsburgh, 223 Tennyson Avenue, Pittsburgh, PA 15213, USA; 2Department of Supportive Oncology, Dana-Farber Cancer Institute, Boston, MA 02215, USA; jian_zhao@dfci.harvard.edu; 3Department of Psychiatry, Harvard Medical School, 25 Shattuck Street, Boston, MA 02115, USA

**Keywords:** medication adherence measurement, self-report, pharmacy refill, electronic monitoring, chronic disease, state-of-the-art review

## Abstract

**Background/Objectives:** One of the most important self-management behaviors is following agreed-upon treatment recommendations. In chronic disease, which affects over one-third of adults, a critical behavior is taking prescribed medication. However, approximately half of patients with chronic conditions fail to adhere to medication recommendations. Research into medication adherence is complicated by the diversity of measurement methods and definitions, resulting in inconsistent outcomes. Accurate measurement is essential for clinical decision making and identifying effective interventions. This state-of-the-art review aimed to map the current landscape of adherence measurement in chronic disease management and provide evidence-based recommendations for future research and practice. **Methods:** Using a state-of-the-art review approach, we examined objective and subjective adherence measures in studies where medication adherence was a primary outcome, published from August 2019 to July 2024. The frequencies of each method type were calculated. In studies using more than one method within a sample, adherence outcomes were compared to assess their comparability. **Results:** Of 1036 screened records, 314 met the inclusion criteria. Self-report questionnaires were most frequently used (72% of studies), followed by pharmacy refill measures (22%), electronic monitoring (2.5%), and biologic assays (1.3%). Subjective measures were more frequently used due to their convenience and lower cost but they reduce the level of precision. Objective measures offered greater precision but at a higher cost and logistical complexity. **Conclusions:** Our findings suggest a dominant reliance on subjective measures. Standardizing definitions, thresholds, and reporting, and adopting multimodal measurement strategies, will improve the validity, comparability, and clinical utility of adherence research.

## 1. Introduction

Chronic diseases, particularly prevalent among older adults, are typically long-lasting (≥1 year), progressive, and managed rather than cured. The most common chronic diseases include cancer, heart disease, stroke, diabetes, and arthritis, which often require long-term pharmacotherapy for effective management. However, inadequate adherence to prescribed medication regimens remains a critical barrier to achieving optimal health outcomes. The World Health Organization (WHO) defines adherence as the extent to which a patient’s behavior aligns with medical recommendations, and poor adherence is widely recognized as a public health challenge, especially for patients with chronic conditions. Consistent with prior summaries, recent estimates suggest that approximately half of all patients fail to adhere to their medication as prescribed, with important implications for preventable morbidity, hospitalizations, and costs [1,2,3]. Improving medication adherence is essential for improving patient outcomes and reducing the overall burden on healthcare systems.

The accurate measurement of medication adherence is vital for both clinical practice and research. In clinical settings, precise adherence measurement helps healthcare providers identify the causes of poor treatment outcomes, while in research, it enhances the reliability of studies on interventions aimed at improving adherence. However, one of the key challenges in adherence research is the significant variability in the methods used to assess adherence, leading to inconsistent outcomes and making it difficult to compare studies directly. The issue of variability in adherence measures leading to variations in reported adherence has been the subject of long-standing discussion, beginning with Gordis’ seminal work [4] and reiterated over a period of more than 40 years [5,6,7,8,9,10,11,12,13,14]. Of primary concern are the discrepancies in reported adherence outcomes based on whether electronic monitoring, pill counts, or self-report measures were used [4,15,16]. Self-reporting has been noted to overestimate adherence when compared with more objective measures [15] and to have independent predictors from monitored adherence [9]. For example, sociodemographic features have been associated with self-reported adherence but not with monitored adherence [9], setting the stage for bias in identifying persons at risk. Further, in a cholesterol-lowering study, electronically monitored adherence was associated with reduced cholesterol while subjective measures were not [8].

Methods for measuring medication adherence can generally be divided into objective and subjective approaches. Objective methods include electronic monitoring, pill counts, pharmacy refill data, and biomarkers, which provide more precise, quantifiable data but tend to be resource-intensive. Although electronic devices, smart packaging, and app-based logs offer granular time stamps, their uptake in routine chronic disease research and care is uneven due to cost, workflow, and the burden of data-management [15,17]. Subjective methods, such as self-reported questionnaires, are more accessible and cost-effective, though they are often less reliable due to potential bias and social desirability biases. Conceptually, the Ascertaining Barriers to Compliance (ABC) taxonomy distinguishes adherence into initiation (first dose), implementation (dose-taking over time), and discontinuation/persistence; different measures capture different phases, helping explain why estimates often diverge across methods [18]. Reporting guidance, such as the ESPACOMP Medication Adherence Reporting Guideline (EMERGE), further recommends that studies specify the adherence phase, data source, and operational definitions to improve comparability [15]. Notably, self-reporting remains the most commonly used approach but tends to overestimate adherence compared with electronic monitoring [19,20].

Accordingly, we conducted a state-of-the-art review [insert reference tGrant & Booth, 2009] [21] to address gaps in the literature by systematically analyzing studies published between 1 August 2019 and 30 July 2024, focusing on the current use of objective and subjective measures of medication adherence in people with chronic diseases. Our objectives were as follows: (1) to map the frequency and types of adherence measures in the identified studies; (2) to summarize operational definitions and thresholds as well as the adherence phase captured; and (3) to compare, where available, adherence estimates within the same samples across multiple measures to assess concordance. By reviewing the latest measurement choices and reporting practices, we aim to determine whether previous recommendations have been addressed and to inform future adherence research and clinical implementation.

## 2. Methods

We conducted a state-of-the-art review, which, by definition, emphasizes synthesizing the most current and influential evidence rather than exhaustively mapping all historical studies. The review focused on the most recent empirical studies, in which medication adherence was the primary outcome, published between August 2019 and July 2024. This deliberate restriction to a five-year window ensured that our synthesis captured the latest advances in adherence measurement methods and reporting practices [19]. Consistent with guidance for state-of-the-art evidence syntheses, this review followed the SALSA framework (search, appraisal, synthesis, analysis): we used transparent database searches and dual-reviewer screening (search), applied a relevance-based appraisal without formal risk-of-bias scoring (appraisal), summarized patterns in measures and thresholds (synthesis), and interpreted implications for validity, comparability, and implementation (analysis) [19]. Our inclusion criteria were deliberately narrow to capture high-relevance research on medication adherence measurement in chronic disease, and we limited our search to studies reporting original quantitative adherence data. Because the aim was to describe and compare measurement approaches, not to synthesize intervention effects or assess causality, the review, while not a traditional systematic review, nevertheless followed transparent and reproducible processes for database searching, study selection, and data extraction, and we reported our findings in alignment with the Preferred Reporting Items for Systematic Reviews and Meta-Analyses (PRISMA) framework where applicable. To enhance transparency of study identification and selection, we present a PRISMA flow diagram; this is a reporting aid and does not indicate a scoping analytical aim for the review.

Eligibility

Participants: Adults (≥18 years) with non-infectious chronic diseases (e.g., cardiovascular disease, diabetes, cancer, chronic respiratory disease, chronic kidney disease). Studies with mixed populations were eligible if data for eligible participants could be extracted separately.Concept: Medication adherence measurement was the primary outcome, including subjective (self-report questionnaires/visual analog scales) and objective methods (pharmacy refill metrics, electronic monitoring, biologic assays).Context: We included studies conducted in any care setting (e.g., primary care, specialty clinics, hospital outpatients, community programs) and published in the English language from 1 August 2019 to 30 July 2024.

Exclusion: We excluded case reports, case series, or studies with fewer than 10 participants (due to limited generalizability); conference abstracts, dissertations, theses, reviews, and editorials (due to lack of complete data); pediatric populations (<18 years); studies where the primary outcome was not medication adherence; and studies examining non-medication regimens only (e.g., diet, exercise, lifestyle modification) without a pharmacologic component.

Information sources and search. With a librarian-informed strategy, we searched PubMed, EMBASE, and Web of Science (1 August 2019–30 July 2024) using the terms “medication adherence” and “chronic disease.” Following JBI’s three-step approach, we (1) ran an initial limited PubMed search to analyze text words/index terms; (2) executed full searches in all databases; and (3) screened the reference lists of the included studies. Full strategies are provided in Appendix A

Selection process. Records were de-duplicated in EndNote. Two reviewers (JZ, JD) independently screened titles/abstracts and full texts; disagreements were resolved by consensus. Reasons for exclusion at full text were recorded and are summarized in the PRISMA-ScR flow.

Data charting. A standardized, pilot-tested charting form was used to extract first author/year, country, disease area, study design, sample size, setting, adherence measurement method (instrument/metric), operational definition/thresholds, and adherence estimate. For cross-sectional studies, we recorded the single adherence estimate reported. For longitudinal or experimental designs, we extracted baseline adherence only to avoid contamination by time or intervention effects. Data were entered into Microsoft Access 2016.

Synthesis. Adherence methods were grouped a priori as subjective (e.g., self-report questionnaires, visual analog scales) or objective (pharmacy refill metrics such as MPR/PDC; electronic monitoring; biologic assays). When studies deviated from canonical thresholds (e.g., PDC ≥ 0.80), the author-defined threshold was recorded verbatim. We summarized the frequency of use for each method category. Within each category, we calculated the mean, median, and range of adherence estimates as reported by the included studies. In multi-method studies, adherence estimates were compared within the same sample to assess comparability across methods. Owing to heterogeneity in instruments, thresholds, and metrics, no meta-analysis was performed; findings are presented descriptively and narratively.

## 3. Results

After reviewing and eliminating duplicates and non-full texts, a total of 314 articles met our selection criteria out of 1036 identified articles. During the screening, we identified 47 papers utilizing qualitative methods to investigate the feelings, difficulties, or barriers to medication adherence, these qualitative-only studies were excluded from quantitative synthesis but informed our contextual interpretation (see Figure 1).

Self-report questionnaires were the most commonly used measurement tool, employed in 225 (72%) studies. The Morisky Medication Adherence Scale (MMAS) was the most frequently used self-report questionnaire (n = 35; 11% of all included studies; 15.6% of reported studies) (Table 1). Many of the identified studies utilized questionnaires adjusted for location, such as the Chinese version of the Medication Adherence Reasons Scale and the Chinese and Western medication adherence scale. Single items were also frequently employed when assessing medication adherence, such as “How many days in the preceding week did you take this medication?”. Self-reported pill counts were used in seven studies. For tools used in ≥4 studies among the 225 self-report papers, we summarize baseline adherence means and ranges in Table 1. The overall mean adherence across commonly used subjective tools (MMAS, ARMS, VAS, MARS, pill count) was approximately 54%, with broad variability by instrument and definitions (see Table 1).

Approximately one-quarter of the studies (n = 69; 22%) utilized pharmacy data to define and calculate adherence, including metrics such as the percentage of days covered (PDC), medication possession ratio (MPR), data from electronic medical records, and national databases. For pharmacy-based measures, the overall mean adherence was 60% (median of 60%; range of 11–98%). Electronic monitoring was used in 19 studies (6.1% of 314 papers) to assess adherence, with an overall mean adherence of 70% (median 74%; range 20–97%). Biological tests were employed in four studies to measure adherence, showing an overall mean of 16% (median of 25%; range 0.3–90%). It should be noted that the association of biological tests is influenced by drug metabolism and responsiveness to drugs, which varies between individuals. Additionally, one paper utilized a clinic-based inquiry to assess adherence, representing a less commonly used approach among the studies reviewed. Across studies, electronic monitoring yielded a higher mean adherence (≈70%) than self-reporting (≈54%). Although counter to expectation, this pattern likely reflects stricter categorization in several SRQ studies (e.g., high cutoffs for “adherent”), heterogeneity in thresholds across instruments, and differences in how continuous vs. categorical metrics were summarized.

In 10 identified studies from 2019 to 2024, multiple measurements that combined subjective and objective measures were utilized. The baseline adherence outcomes among studies using multiple measures are presented in Table 2 (study characteristics and definitions) and Table 3 (side-by-side adherence estimates) [26,27,28,29,30,31,32,33,34,35]. The studies included in the table utilized different medications and measurement methods, including electronic monitoring, medication adherence questionnaires, pill counts, self-reporting, proportions of days covered, and biological tests. Across these within-sample comparisons, three regularities emerge. First, refill metrics often exceed EM (e.g., Hesso: PDC 97.8% and MRA 100% vs. EM 42.7% [27]; Mehas: MPR 89–93% vs. cap-based EM 55–61% [32]). Second, self-reports sometimes exceed EM (Pericot-Valverde: VAS 94.9% vs. blister 78.2% [29]; Arutyunov: inquiry 89.0% vs. EM 34.4% [28]), yet the reverse also occurs (Tangirala: EM 70–71% vs. MARS-10 38–39% [30]; Boons: EM/pill count ~79–80% vs. MARS-5 68% [26]). Third, biologic assays tend to be lowest in some contexts (Mielke: mean self-report 16.6% vs. mean urine assay 8.74%) [33], while alignment is possible in others (Thompson: self-report > 96% and HPLC-MS/MS > 90%) [34]. Together, these contrasts make clear that measurement choice and its operational definition (e.g., thresholds, windows, continuous vs. categorical scoring) materially shape the proportion classified as “adherent.” This observation helps explain our summary findings (adherence was approximately 70% by electronic monitoring vs. 54% by self-report).

## 4. Discussion

Over 70% of the identified papers used self-reported questionnaires, making it the most commonly used measure for assessing medication adherence. Pharmacy data were used in nearly 22% of the studies, while objective measures, including electronic monitoring and biologic tests, were used in only approximately 7% (23/314) of the studies. In our sample, electronic monitoring yielded the highest mean adherence of about 70% among all measures, compared with 54% for self-reporting and 60% for pharmacy refill-based metrics. This counterintuitive SRQ-vs-EM difference reinforces that feasibility-driven choices (SRQ) can trade precision for accessibility and underscores that measurement choice and its operational definition shape the adherence estimate. Although the absolute means differ by method, all are below levels typically associated with optimal control of chronic disease.

Despite the expectation that objective measures of medication adherence should be used more often, self-reported questionnaires (SRQs) were the most commonly used measure, particularly in cross-sectional studies. As previously reported, there are numerous adherence scales available [15]. These adherence scales include items aimed at assessing patients’ medication-taking behaviors, such as experiential feedback and habit strength, as well as identifying barriers to good medication-taking behaviors or beliefs. While SRQs capture behaviors and beliefs related to taking and refilling medicines, composite scoring makes it difficult to isolate “tablets consumed” from attitudes or barriers. Several studies comparing SRQs with electronic monitoring or refill records show that SRQs tend to overestimate adherence relative to objective measures [20,36]. However, our lower SRQ mean (54%) likely reflects heterogeneity in instrument cutoffs and stricter categorization rules in several included studies, as well as well-known limits of episodic recall [37]. This pattern reflects a trade-off between feasibility and precision. The dominance of self-reporting prioritizes access and cost but may compromise validity, which might further affect how we judge intervention effects and identify at-risk patients.

The Morisky Medication Adherence Scale (MMAS) was the most commonly used self-reported questionnaire to measure adherence. While the four-item version (MMAS-4) includes elements of forgetfulness and symptom severity, the eight-question version (MMAS-8) delves into other situational and emotional aspects of medication adherence [22]. For instance, the eight-question version also assesses non-adherence due to feelings of pressure or reasons other than forgetfulness [22]. The definition of adherence when using MMAS varies across studies. For MMAS-8, a score of less than six is often categorized as low adherence, six to seven as medium adherence, and eight as high adherence, although some studies have exceptions. For example, in a study exploring antihypertensive medication adherence, an MMAS-8 score of six or higher was declared adherence. The scoring for MMAS-4 is opposite, with 0 indicating high adherence, one to two as medium adherence, and greater than two as low adherence. Some studies also adapted MMAS content, for example, by adding preference-related items [38]. The mean adherence from selected MMAS studies was 62.5% (range: 38.3–92.3%). Importantly, MMAS responses do not directly verify the number of pills taken or the timing of intake, meaning that high reported adherence may not correspond to correct dosing behavior.

In addition to MMAS, the Adherence to Refills and Medication Scale (ARMS) [23] and the Medication Adherence Report Scale (MARS) [24] are also frequently utilized to measure medication adherence. The ARMS assesses both medication taking and timely refilling, and the MARS-5 identifies specific nonadherence behaviors such as forgetting or deliberately missing doses. These questionnaires demonstrate moderate validity compared with objective measures, with an ARMS sensitivity/specificity of 61%/78% and a MARS sensitivity/specificity of 77%/64%. However, definitions and cutoffs vary (e.g., MARS-5 cutoff values range from 20 to 25; VAS cutoffs range from 80% to 95%), which limits comparability [39,40].

Objective methods for measuring medication adherence vary in their accuracy, feasibility, and interpretability. Pill count remains a widely used approach, directly quantifying remaining doses; however, it cannot verify ingestion, timing, or dosing frequency, and accuracy depends on the assumption that missing pills were consumed rather than lost or discarded. When pill counts are supplemented with self-reported missed-dose recall, additional biases are introduced, as recall periods and reporting formats differ across studies, making comparisons challenging. Pharmacy refill data—typically calculated as medication possession ratio (MPR) or proportion of days covered (PDC)—were used in about one-quarter of studies in this review (69/314). It should be noted that this is a measure of refilling medication prescriptions and not of medication that is taken. MPR and PDC yield similar results for a single drug class but differ in handling overlapping prescriptions. Most studies applied the conventional ≥0.80 cutoff for adherence, though thresholds up to 0.95 were reported, and some used continuous values without a predefined cutoff. Adherence estimates ranged widely (36–89.4%), reflecting differences in populations, regimens, and operational definitions. Refill data offer scalability and a low cost but, like pill counts, do not confirm ingestion or capture day-to-day dose timing.

Electronic monitoring (EM) systems, such as the Medication Event Monitoring System (MEMS), were less common in our sample (6.1% of studies) but yielded the highest mean adherence (70%). EM records the date and time of container openings, providing granular dosing data; newer modalities include electronic blister packs, connected pill dispensers, and app-based logging. Although EM is often cited as the “gold standard” for implementation-phase adherence [18], its use is limited by device cost, data management demands, and the potential for device-triggered behavior change (Hawthorne effect) [36]. In our review, SRQ adherence means were unexpectedly lower than those for EM and refill data, possibly due to stricter adherence classification criteria in several included SRQ studies compared with the dichotomous 80% threshold used for many objective measures. Recent reviews emphasize that no single objective method is without limitations [39]. The choice of measure should align with the adherence phase (initiation, implementation, persistence) targeted, be reported according to EMERGE guidelines, and, where possible, be combined with complementary methods to improve validity and contextual interpretation [41].

Based on the above analysis, we can better understand why so little has changed despite repeated recommendations. Several system-level and methodological factors help explain the slow progress. First, feasibility and workflow constraints favor self-reporting in busy clinics and survey-based studies (low cost, minimal infrastructure), whereas electronic monitoring requires device procurement, data management, and participant training. Second, misaligned incentives persist between payers and health systems, which frequently track refill-based metrics for population management, reinforcing reliance on possession, rather than ingestion, indicators. Third, heterogeneous operational definitions and cutoffs across instruments (e.g., study-specific SRQ thresholds; varying PDC/MPR rules) make harmonization difficult and limit the perceived payoff of switching methods. Fourth, taxonomy/reporting guidance is under-utilized in practice: studies often do not specify the adherence phase assessed, observation window, or handling of overlaps, blunting comparability even when similar tools are used. Together, these factors create methodological inertia—a pattern also reflected in our sample, where self-reporting dominated use but produced a lower mean adherence than EM, underscoring how operational choices shape estimates.

Our findings are consistent with prior reviews documenting substantial heterogeneity in adherence measures and the absence of a universal gold standard. They also echo calls for standardization and transparent reporting [12,39]. The within-sample discrepancies we summarize are consistent with work comparing MEMS against self-reporting and refill methods [36]. Our five-year snapshot extends the current literature by quantifying recent method use and highlighting where multimodal strategies and common thresholds would improve validity and comparability [39].

As noted, a large variety of measures have been and continue to be used in ongoing adherence research. A comparative analysis of various tools designed to assess medication adherence is presented in Table 4 which highlights how each tool aligns with the World Health Organization’s (WHO) central emphasis on medication adherence while also exploring the broader context of medication adherence. For instance, the MMAS evaluates a patient’s medication-taking routine and reasons for missing doses, with its extended eight-item version further delving into emotional and situational reasons for non-adherence. The ARMS examines both the act of taking the medication and the timeliness of refilling prescriptions, emphasizing the logistical aspects of adherence. Tools like electronic medication monitoring and pharmacy refill records provide empirical data, either through monitoring medication bottle openings or tracking prescription refills, respectively. Beyond just confirming if medications are taken, these tools collectively capture the multifaceted nature of medication adherence, considering factors like patient beliefs, accessibility issues, and physiological responses. In ABC terms, most self-report scales and EM primarily assess the implementation phase (dose-taking over time), whereas refill metrics primarily index persistence (continuation over an interval) and biologic assays confirm ingestion within a pharmacokinetic window. These phase differences, compounded by non-uniform cutoffs and observation windows, can predict the cross-method discrepancies we observed in the multi-method studies (Table 2 and Table 3).

It is important to acknowledge that there is no universally accepted gold standard measure for medication adherence. Investigators and clinicians are encouraged to be clear about the characteristics of adherence they aim to measure and to report the nature of the assessment undertaken with the chosen measure. To address the limitations of individual measures, particularly in adherence research, a combined assessment using both subjective and objective measures is recommended. Future research should aim to explore and utilize measures that improve the accuracy and completeness of medication adherence assessment.

Implications for practice and research: Given the predominance of self-report measures and the variability in cutoffs across studies [12], we would recommend pairing brief self-report screening within clinics with available refill data (e.g., PDC/MPR) and consider targeted, time-limited objective checks (e.g., electronic monitoring) when results are equivocal or patient risk is high [21,36]. In practice, adherence measure selection should be aligned to the ABC phases (initiation, implementation, persistence) [18] and documentation should specify the phase, data source, threshold, and operational definition in line with EMERGE guidance [2]. In research studies, we would recommend measures that best approximate patients’ medication-taking behaviors, focusing on the percentage of prescribed doses taken, again following the EMERGE guidance. At the policy level, health systems can improve comparability by adopting standardized definitions and thresholds across service lines [18], integrating pharmacy/EHR data to generate adherence dashboards and supporting reimbursement for personnel time and objective monitoring when warranted [14,21]. To operationalize these recommendations, Figure 2 presents a practical framework that integrates the ABC taxonomy with EMERGE reporting elements. Panel A specifies a low-burden minimum reporting set (phase, data source/metric, threshold with justification, observation window/overlaps, and a concordance plan) that can be reported as a single paragraph to immediately improve comparability and reproducibility. Panel B outlines scenario-based multimodal combinations that balance feasibility and precision across common use cases (primary care, high-stakes oral therapies, inhaled therapies, and population surveillance).

## 5. Conclusions

In conclusion, measuring treatment adherence accurately is crucial for identifying problems with adherence. It is also crucial for providing feedback to patients and identifying evidence-based strategies to improve adherence. Patients cannot self-manage their medications if the data are inaccurate or if interventions are based upon studies with questionable outcome measures. Further, investigators cannot report their outcomes with confidence, nor can systematic reviews be conducted when outcomes are variable and biased by measurement methodology.

This review found that despite multiple recommendations that the definition of adherence be standardized and that the measures selected be valid, little change has been seen over time in either the definition or measurement of adherence. There is a wide range of adherence measurements utilized, with varying definitions and cutoff values, with little validation data. While simple subjective measures were the most commonly used, big data sets like pharmacy dispensing databases and national health system records provide a unique opportunity to study adherence at the population level, recognizing that these measures assess the availability of medication and not the actual taking of medication. However, standardization is lacking, and a more comprehensive objective measure of patient adherence is needed. A combination of objective and subjective measures would be optimal to determine actual adherence rates, as defined by the WHO, and gather information on patient perceptions and circumstances surrounding adherence events. The field needs to evaluate the construct validity of these multiple measures and identify accurate measures of the percentage of medication taken and patterns of medication adherence, utilizing standard definitions of adherence. Variability in measures, definitions, and calculation strategies continues to compromise progress in understanding and improving patient adherence.

## Figures and Tables

**Figure 1 nursrep-15-00370-f001:**
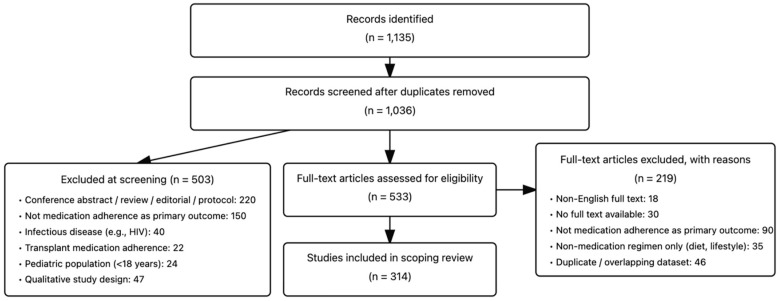
PRISMA-ScR flow diagram.

**Figure 2 nursrep-15-00370-f002:**
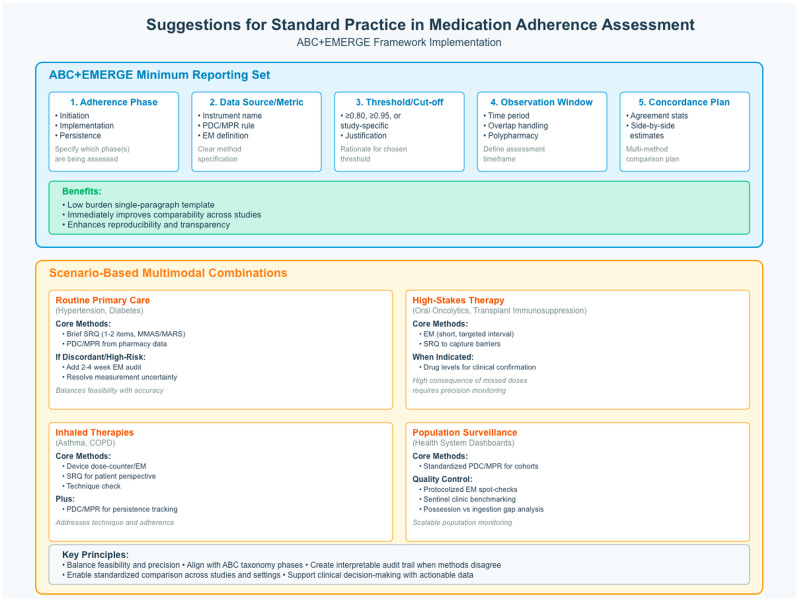
Suggestions for standard practice in medication adherence assessment: ABC+EMERGE minimum reporting set and scenario-based multimodal combinations.

**Table 1 nursrep-15-00370-t001:** The use of popular medication adherence tools and adherence.

	Instrument/Metric	Studies (n)	Operational Adherence Definition Used in Included Studies	Reported Adherence Range (% Adherent Unless Notes)	Pooled Mean Adherence (% Adherent)	Pooled Median Adherence (% Adherent)
Subjective measures	Morisky Medication Adherence Scale (MMAS) (4/8-item)	35	Proportion of participants with a score above 6 in MMAS-8 or a score of 0 in MMAS-4	38.33–93.3%	62.5%	61%
	Adherence to Refills and Medication Scale (ARMS)	6	Proportion of participants meeting study-defined ARMS threshold (lower score = better adherence; typical cutoffs ≤ 12–16)	42.7–85.3%	57.6%	53%
	Visual Analog Scale (VAS)	4	Proportion reporting adherence ≥ study cutoff (commonly 95%)	42% to 73%	46%	49%
	Medication Adherence Report Scale (MARS)	6	Proportion of participants with a score above 20–25 (cutoff value of adherence ranges from studies) in MARS	7% to 84%	30%	16%
	Pill count	7	Proportion of participants who did not miss certain doses in a recall period	53–92%	75%	72%
Objective measures	Pharmacy refill (MPR, PDC)	69	Proportion of participants with PDC or MPR ≥ study cutoff (commonly 0.80–0.95)	11% to 98%	60%	60%
	Electronic monitoring	19	Proportion of participants above a specific ratio, which is usually calculated based on the number of observed cap openings divided by the number of prescribed doses per day in a specific period	20% to 97%	70%	74%
	Biological test	4	Proportion with drug/metabolite detected within assay-defined threshold	0.3% to 90%	16%	25%

Note: Psychometric support for subjective tools includes MMAS meta-analytic reliability and limited criterion validity at the common cutoff (α ≈ 0.67–0.77; ICC ≈ 0.80; sensitivity 0.43, specificity 0.73) [MMAS-8] [20,22], ARMS internal consistency and clinical/behavioral validity (α ≈ 0.81) [ARMS] [23], MARS-5/10 reliability, test–retest and construct/criterion validity [MARS] [24], and VAS convergence with MEMS/pill count and meta-analytic support [25].

**Table 2 nursrep-15-00370-t002:** Summary of papers using both objective and subjective measures of adherence.

Author	Type of Study	Population	Medication	Adherence Measurement	Definition	**Adherence** **Outcomes**
(Boons et al., 2020) [26]	Observational study	Chronic myeloid leukemia patients	Nilotinib	MEMS(%PDC)	Proportion of participants with PDC > 95%	78.7%
				Pill count(%AR) *	Proportion of participants with AR > 95%	80%
				Medication Adherence Report Scale (MARS-5)	Proportion of participants with MARS scores above 25 at 12 months	68%
(Arutyunov et al., 2022a) [28]	Observational study	Post-MI patients or patients with hypertriglyceridemia	Omega-3-acid ethyl esters (OM3EE)	Electronic monitoring	Proportion of participants with ratio ≥ 0.8 (The ratio of days when the full prescribed dose of OM3EE was taken to the total number of days in the treatment period)	34.4%
				Inquiry at clinic visits	Proportion of participants with scores above 8 in Questionnaire of Treatment Compliance	89.0%
(Hesso et al., 2020) [27]	Observational study	Patients with COPD or asthma	Inhaler technique	Electronic monitoring	Proportion of participants with dose counter ≥ 80%	42.7%
				Medication refill adherence (MRA)	Proportion of participants with MRA ≥ 80%	100%
				Proportion of days covered (PDC)	Proportion of participants with PDC ≥ 80%	97.8%
				Self-reporting—inhaler adherence scale (IAS)	Proportion of participants with scores ≥ 4 in IAS	48%
(Pericot-Valverde et al., 2021) [29]	Secondary analysis of an RCT	Patients with HCV	Direct-acting antivirals (DAAs)	Electronic blister packs	Average blister pack adherence of the total sample	78.2%
				Self-reported—VAS	Average self-reported VAS (0–100%) adherence of the total sample	94.9%
(Tangirala et al., 2020) [30]	Cross-sectional study	Patients with COPD or asthma	Inhaled corticosteroids (ICS), long-acting beta-agonists (LABA), and long-acting anti-muscarinic agents (LAMA)	Electronic monitoring devices and dose counts from analog counters	Proportion of participants with ≥80% of total doses prescribed, per convention	70% for asthma and 71% for COPD
				Self-reported—10-item Medication Adherence Report Scale (MARS) (MARS assessment was defined as a score ≥ 4.5)	Proportion of participants with MARS-10 scores above 4.5	38% for asthma and 39.2% for COPD
(Daud et al., 2021) [31]	Quasi-experimental study	Patients with hypertension	Medication for hypertension	Medication Adherence Questionnaire (MAQ)	Proportion of participants with MAQ scores below or equal to 2	89.6% on counseling group and 86.2% at non counseling group at baseline
				Pill count	Proportion of participants with ≥80% pill count adherence scores	55.2% on counseling group and 65.5% at non counseling group at baseline
(Mehas et al., 2021) [32]	Randomized pilot study	Patients with hypertension	Medication for hypertension	Medication possession ratio	Average proportion of days that a patient had blood pressure medication available to take during the 6 months prior to study enrollment	92.7% (control group) 89.1% (intervention group)
				Electronic cap adherence	This percentage was averaged across all 6 months for each medication	55.1% (control group) 60.7% (intervention group)
				MMAS-8	(Range 0–8) (Mean ± SD)	5.7 ± 1.3 (control group) 6.2 ± 1.1 (intervention group)
(Mielke et al., 2022) [33]	Prospective cohort study	Patients with chronic kidney disease	Antihypertension medications	Self-reported medication	Average self-reported dose rate	Ranged from 1.1% (methotrexate) to 55% (β-blockers) with mean 16.6%
				Biological test	Detectable urinary drug metabolites measured by mass spectroscopy	ranged from 0.3% (3-hydroxyvalproate) to 58% (4-acetaminophen sulfate) with mean 8.74%
(Thompson et al., 2021) [34]	A cohort study	Patients with stable angina	Cardiovascular medication	Self-reported	Average self-reported dose rate	>96% for all drugs
				Biological test	High-performance liquid chromatography with tandem mass spectrometry (HPLC MS/MS)	>90% for all drugs
(Arutyunov et al., 2022b) [28]	A prospective cohort study	Patients with a history of recent myocardial infarction or endogenous hypertriglyceridemia	Omega-3 polyunsaturated fatty acids	National Questionnaire of Treatment Compliance (NQTC)	Proportion of participants with NQTC scores equal to or higher than 8	87.9% in post-MI group and 90.2% in hypertriglyceridemia group
				Digital adherence monitoring system (DIAPASon)	The total number of days that a patient took the full prescribed dose of OM3EE during the specified period divided by the total number of days in that period	37%
(Calvo-Arbeloa et al., 2020) [35]	A cross-sectional observational analysis	Patients with inflammatory bowel disease	Adalimumab, golimumab and ustekinumab	Medication possession ratio	Proportion of patients with MPR ≥ 85%	74.7%
				Morisky–Green Medication Adherence Questionnaire (MMAS-8)	Proportion of patients with MMAS ≥ 6	Only assessed for MPR non-adherent patients: 53.1%

*: Adherence rate.

**Table 3 nursrep-15-00370-t003:** Adherence outcome for studies using multiple measures of adherence.

	Pill Count	PDC/MPR	Electronic Monitoring	Self-Reported	Biological Test
(Boons et al., 2020) [26]	80%	78.7%		68%	
(Hesso et al., 2020) [27]		97.8%	42.7%	100% for MRA and 48% for IAS	
(Pericot-Valverde et al., 2021) [29]			78.2%	94.9%	
(Tangirala et al., 2020) [30]			70% for asthma and 71% for COPD	38% for asthma and 39.2% for COPD	
(Daud et al., 2021) [31]	55.2% in counseling group and 65.5% in non-counseling group at baseline			89.6% in counseling group and 86.2% in non-counseling group at baseline	
(Mehas et al., 2021) [32]		92.7% (control group) 89.1% (intervention group)	55.1% (control group) 60.7% (intervention group)	5.7 ± 1.3 (control group) 6.2 ± 1.1 (intervention group)	
(Mielke et al., 2022) [33]				Ranged from 1.1% (methotrexate) to 55% (β-blockers) with mean 16.6%	Ranged from 0.3% (3-hydroxyvalproate) to 58% (4-acetaminophen sulfate) with mean 8.74%
(Thompson et al., 2021) [34]				>96% for all drugs	>90% for all drugs
(Arutyunov et al., 2022b) [28]			37%	87.9% in post-MI group and 90.2% in hypertriglyceridemia group	
(Calvo-Arbeloa et al., 2020) [35]		74.7%		53.1% (for non-adherent pts based on MPR criteria)	

**Table 4 nursrep-15-00370-t004:** Commonly used medication adherence measures, their alignment with WHO’s definition and additional considerations.

Tool	Alignment with WHO Definition	ABC Phase Best Captured *	Additional Considerations
MMAS (4-/8-item)	Evaluates the act of taking medication and reasons for missed doses	Implementation	Quick, widely used; cutoffs vary, and ceiling effects occur; reflects beliefs as well as behavior → may not equal tablets consumed. Recent reviews emphasize heterogeneity and self-report bias.
ARMS	Assesses both medication taking and timely prescription refills.	Implementation → Persistence	Valid in chronic disease and low-literacy groups; captures refill barriers. Thresholds vary across studies.
MARS-5	Focuses on specific non-adherence behaviors (e.g., forgetting, intentional omission).	Implementation	Distinguishes intentional vs. unintentional non-adherence; cutoffs differ (e.g., 20–25). New psychometric work continues to appear.
Electronic medication monitoring (e.g., MEMS; smart blisters/boxes, app logs)	Directly records date/time of container openings (proxy for dose taking).	Implementation → Persistence	High temporal granularity (patterns, weekends, timing); does not confirm ingestion; device cost/data burden; possible Hawthorne effect. Recent systematic reviews (incl. DOACs; digital EAM) underscore value but variable implementation/acceptability
Pharmacy refill records (PDC/MPR)	Compares expected vs. actual remaining doses.	Persistence → Implementation	Scalable/low cost; cannot confirm ingestion; thresholds vary (often ≥ 0.80, sometimes ≥ 0.95). Reporting per EMERGE improves comparability
Pill count	Detects drug/metabolite levels to confirm ingestion.	Implementation	Low cost and simple; vulnerable to pill dumping/loss; no timing info; best when paired with another method.
Biological tests	Drug/metabolite levels (evidence of recent ingestion).	Initiation/Implementation	Confirms exposure but depends on PK half-life, assay, sampling timing; costly and less feasible for routine use

ABC phase best captured * ABC phases = initiation (first dose), implementation (dose-taking over time), discontinuation/persistence (when therapy stops/continues). Abbreviations: WHO = World Health Organization; ABC = Ascertaining Barriers to Compliance taxonomy; MMAS = Morisky Medication Adherence Scale; ARMS = Adherence to Refills and Medications Scale; MARS-5 = 5-item Medication Adherence Report Scale; MEMS = Medication Event Monitoring System; PDC = Proportion of Days Covered; MPR = Medication Possession Ratio; PK = Pharmacokinetics; EMERGE = ESPACOMP Medication Adherence Reporting Guideline for Evidence-based Recommendations.

## Data Availability

No new data were created.

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
