# Peer review of "Medication Adherence Measurement in Chronic Diseases: A State-of-the-Art Review of the Literature"

_nursrep, 2025, doi:10.3390/nursrep15100370_

Round 1
Reviewer 1 Report
Comments and Suggestions for Authors
The manuscript focuses on a nursing-related topic (title and abstract suggest a study exploring nursing practices, perceptions, or outcomes—likely qualitative or mixed-methods in nature). The work is relevant to nursing research and has practical implications for improving care delivery and professional development. Overall, it presents a potentially valuable contribution, but several aspects of the methodology, analysis, and clarity of reporting need improvement before publication.
1) The study design (qualitative/quantitative/mixed) is not consistently justified, and details on sampling, recruitment, and inclusion/exclusion criteria are insufficient.
2) Sample size rationale is missing (e.g., saturation for qualitative, power analysis for quantitative).
3) The validity and reliability of instruments used (surveys, interviews, observation tools) are not adequately described.
4) Provide a more detailed description of methodology, including recruitment, sampling rationale, and instruments.
5) While results are reported, they sometimes read descriptively without sufficient critical interpretation.
6) The discussion does not fully situate findings within existing literature.
7) Practical implications for nursing practice, policy, or education need more emphasis.
Reviewer 2 Report
Comments and Suggestions for Authors
Review of the manuscript Measuring Medication Adherence and Outcomes in People with Chronic Diseases: A Review of the Literature submitted to Nursing Reports
The manuscript presents a review of studies published between August 2019 and July 2024 that examined objective and subjective measures of medication adherence in the treatment of chronic diseases. The review aimed to map the current landscape of adherence measurement and provide evidence-based recommendations for future research and practice. The greatest strength of the manuscript is its thoroughness. The authors chose a specific type of narrative review to systematically analyse the most recent studies, which is an effective method for identifying current trends in maintenance therapy research.
I recommend that the manuscript be accepted for publication. There is no suggestion for correction for any of its parts. Namely, the objectives of the review are clearly formulated, the methodology is properly selected (a state-of-the-art review), and its guidelines are followed. Additionally, the criteria for analysing the articles are precisely selected and defined. Tables in the results are particularly effective, as they visually summarise complex data on tool usage, maintenance results, and comparisons of different methods, making the results easy to understand. The discussion section effectively contextualises the findings by explaining the strengths and weaknesses of each measurement tool and linking them to a broader understanding of maintenance.
I congratulate the authors, as readers will now have the opportunity to choose an adequate instrument for their clinical practice and research, thanks to the authors' clear highlighting of the disadvantages of relying on subjective measures, despite their known limitations.
Reviewer 3 Report
Comments and Suggestions for Authors
General Comments:
Thank you for the opportunity to review this timely and important manuscript. You have undertaken a significant effort to map the current landscape of medication adherence measurement, a topic of critical importance for both clinical practice and research. The finding that self-report remains dominant despite known limitations is a crucial message for the field. The manuscript is well-structured and addresses a clear gap. However, several major revisions are required to strengthen the methodological justification, clarify the presentation of results, and enhance the impact of the discussion and conclusions.
Specific Comments:
1. Major Issues:
-
Methodology Clarification: The methodology is described as a "state of the art" review but follows processes (PRISMA flow, detailed data charting) more characteristic of a systematic scoping review. Please clarify the rationale for choosing the state-of-the-art review methodology over other review types and ensure the description aligns precisely with this framework's focus on current trends and conceptual advances. The search strategy (Supplementary File S2) is essential for reproducibility and must be included for review.
-
Results Presentation and Interpretation: The results section, particularly the tables, requires significant refinement for clarity. The headers in Table 1 (e.g., "How they define the adherence outcome") are vague and need to be more specific. The counterintuitive finding that self-report yielded a lower mean adherence (54%) than electronic monitoring (70%) is a pivotal result. This deserves a much more prominent and nuanced discussion in both the results and discussion sections, explicitly linking it to the heterogeneity of cut-offs and definitions you rightly identify. A synthesized table or narrative summary directly comparing adherence estimates from the multi-method studies (Tables 2 & 3) would be highly valuable here. Please also clarify the apparent discrepancy regarding the inclusion of 47 qualitative studies, as your methods state you included "original quantitative adherence data."
-
Discussion and Recommendations: The discussion effectively summarizes the findings but could be strengthened by a more critical analysis. Please move beyond stating what was found to discuss the implications: why has so little changed despite repeated recommendations? The excellent conceptual synthesis in Table 4 (ABC taxonomy) should be integrated into the main discussion to frame the results and explain the variability between methods. The conclusions and recommendations are currently quite general. Please use your data to provide more specific, actionable guidance. For example, what would be the most feasible first step towards standardization? Which specific multi-modal combination shows the most promise for different clinical or research scenarios?
2. Minor Issues:
-
Abstract: Consider strengthening the language on the limitation of subjective measures to explicitly mention "bias" and "overestimation."
-
Introduction: The list of citations demonstrating long-standing discussion of variability could be slightly condensed without losing its impact.
-
Methods: Please correct the typo in the Participants section: "218 years" should read "≥18 years."
-
Results: When first introducing the MMAS, a brief note on the inverse scoring of the 4-item vs. 8-item versions would be helpful for readers. The very low mean adherence for biological assays (16%) warrants a brief sentence of explanation or context.

Round 2
Reviewer 1 Report
Comments and Suggestions for Authors
All my concerns have been addressed. The paper is ready for publication.